# Using a Double-Core Structure to Reduce the LUT Count in FPGA-Based Mealy FSMs

Alexander Barkalov [1,2,*] , Larysa Titarenko [1,3] and Kazimierz Krzywicki [4,*]

1   Institute of Metrology, Electronics and Computer Science, University of Zielona Gora, ul. Licealna 9, 65-417 Zielona Gora, Poland
2   Department of Computer Science and Information Technology, Vasyl Stus' Donetsk National University (in Vinnytsia), 600-Richya Str. 21, 21021 Vinnytsia, Ukraine
3   Department of Infocommunication Engineering, Faculty of Infocommunications, Kharkiv National University of Radio Electronics, Nauky Avenue 14, 61166 Kharkiv, Ukraine
4   Department of Technology, The Jacob of Paradies University, ul. Teatralna 25, 66-400 Gorzow Wielkopolski, Poland
*   Correspondence: a.barkalov@imei.uz.zgora.pl (A.B.); kkrzywicki@ajp.edu.pl (K.K.)

**Abstract:** A method is proposed which aims at reducing the numbers of look-up table (LUT) elements in logic circuits of Mealy finite state machines (FSMs). The FSMs with twofold state assignment are discussed. The reduction is achieved due to using two cores of LUTs for generating partial Boolean functions. One core is based on maximum binary state codes. The second core uses extended state codes. Such an approach allows reducing the number of LUTs in the block of state codes' transformation. The proposed approach leads to LUT-based Mealy FSM circuits having three levels of logic blocks. Each partial function for any core is represented by a single-LUT circuit. A formal method is proposed for redistribution of states between these cores. An example of synthesis is shown to explain peculiarities of the proposed method. An example of state redistribution is given. The results of experiments conducted with standard benchmarks show that the double-core approach produces LUT-based FSM circuits with better area-temporal characteristics than they are for circuits produced by other investigated methods (Auto and One-hot of Vivado, JEDI, and twofold state assignment). Both the LUT counts and maximum operating frequencies are improved. The gain in LUT counts varies from 5.74% to 36.92%, and the gain in frequency varies from 5.42% to 12.4%. These improvements are connected with a very small growth of the power consumption (less than 1%). The advantages of the proposed approach increase as the number of FSM inputs and states increases.

**Keywords:** Mealy FSM; FPGA; LUT; synthesis; core; twofold state assignment; extended state codes



## 1. Introduction

Our time is characterized by the widespread penetration of various embedded systems into all spheres of human activity [1–3]. Various sequential devices are an integral part of almost every embedded system [4,5]. Very often, the behaviour of a sequential device is represented using the model of Mealy finite state machine (FSM) [6,7]. Often in the FSM design process, designers strive to balance the values of the three main characteristics of a resulting circuit [8,9]. These characteristics are the occupied chip area, maximum operating frequency, and power consumption. The values of these characteristics are closely related [10]. As a rule, the occupied chip area has the greatest influence on the values of other characteristics [11]. The occupied chip area can be reduced using methods of structural decomposition [11]. One of these methods is a method of twofold state assignment (TSA) leading to three-level FSM circuits [12]. The TSA is aimed at Mealy FSMs implemented with field-programmable gate arrays (FPGAs) [13–17].

We chose FPGAs as the basis for the implementation of FSM circuits, since they are widely used for designing various digital systems [18]. We discuss FSM circuits

based on configurable logic blocks (CLBs) consisting of look-up table (LUT) elements and programmable flip-flops. Now, the largest manufacturer of FPGA chips is AMD Xilinx [19]. Due to it, we focus this paper on FPGAs of AMD Xilinx. We propose a method of reducing the numbers of LUTs (LUT counts) in the FPGA-based circuits of Mealy FSMs.

The main disadvantage of twofold FSMs is the need to convert all maximum binary state codes (MBCs) into so-called extended state codes (ESCs) [12]. For this purpose, an additional block is used to transform the maximum binary state codes into the extended state codes. This block consumes some of the FPGA chip's internal resources (LUTs and programmable interconnections). In this paper, we propose a method which allows reducing the overhead connected with the transformation of state codes.

The main contribution of this paper is a novel design method aimed at reducing the LUT counts in the circuits of FPGA-based Mealy FSMs with twofold state assignment. We propose to represent an FSM circuit as a double-core structure. The first core uses maximum binary state codes for generating partial Boolean functions (PBFs). The PBFs of the second core are based on the extended state codes. The proposed approach leads to a LUT-based Mealy FSM where only a part of maximum binary state codes is transformed into extended state codes. Our current research shows that this approach leads to FSM circuits having fewer LUTs compared to FSM circuits based on the twofold state assignment. The experimental results show that FSMs based on our method have practically the same values of the maximum operating frequencies as they are for equivalent FSMs with TSA.

The further text of the article is organized in the following order. The second section shows the background LUT-based Mealy FSM design. The third section discusses the relative works. The main idea of the proposed method is shown in the fourth section. The fifth section includes an example of FSM synthesis using our approach. An algorithm of state redistribution is shown in the sixth section. The seventh section is devoted to results of experiments. The article also includes a short conclusion.

## 2. Background of LUT-Based Mealy FSMs

A Mealy FSM is characterized by sets of states $A$, inputs $X$, outputs $Y$, state variables $T$, and input memory functions (IMFs) $D$ [6]. These sets are the following: $A = \{a_1, \ldots, a_M\}$, $X = \{x_1, \ldots, x_L\}$, $Y = \{y_1, \ldots, y_N\}$, $T = \{T_1, \ldots, T_R\}$, and $D = \{D_1, \ldots, D_R\}$. So, a Mealy FSM has M states, L inputs, N outputs, R state variables and R input memory functions. The values of the first three parameters are independent of the FSM circuit designer. The value of R can be chosen by a designer. The minimum value of R is determined as

$$R_{MB} = \lceil log_2 M \rceil. \tag{1}$$

The Formula (1) determines so-called maximum binary state assignment. The maximum value of R corresponds to so-called one-hot state assignment: $R_{OH} = M$ [20].

The state variables $T_r \in T$ are used for creating state codes $K(a_m)$. An input memory function $D_r \in D$ can set up the binary value of the $r$-th bit of the code $K(a_m)$. To keep state codes, a special register RG is used. The RG consists on R flip-flops controlled by two pulses, *Start* and *Clock* [21]. The pulse *Start* loads the code $K(a_1)$ of the initial state $a_1 \in A$ into RG. The synchronization pulse *Clock* allows loading a state code into RG. This code is determined by the values of IMFs. We discuss a case when the RG consists of flip-flops with informational inputs of D type. This is the most popular type of flip-flops using in the FPGA-based design [18].

In this article, we discuss a case when the internal resources of an FPGA chip are used for implementing FSM circuits. These resources include LUTs, flip-flops, programmable interconnections, synchronization tree, programmable input-outputs [22,23]. The LUTs and flip-flops are combined into CLBs.

A LUT is a block having $S_L$ inputs and a single output [20,24]. A LUT may implement an arbitrary Boolean function including no more than $S_L$ arguments. The value of $S_L$ is rather small [22]. If the number of arguments of a Boolean function exceeds $S_L$, then it is necessary to combine together some LUTs. It is quite possible that a function is represented

by a multi-CLB circuit. In this case, it is necessary to diminish the number of LUTs and their levels in the corresponding circuit [25,26]. In this article we use the symbol LUTer to show that a corresponding logic blocks includes LUTs, flip-flops and interconnections.

An FSM logic circuit is represented by the following systems of Boolean functions (SBFs) [9]:

$$D = D(T, X);\qquad(2)$$

$$Y = Y(T, X).\qquad(3)$$

The SBF (2) represents the function of transitions, the SBF (3) represents the function of outputs [6]. The SBFs (2) and (3) represent a structural diagram of $P$ Mealy FSM (Figure 1) [6].

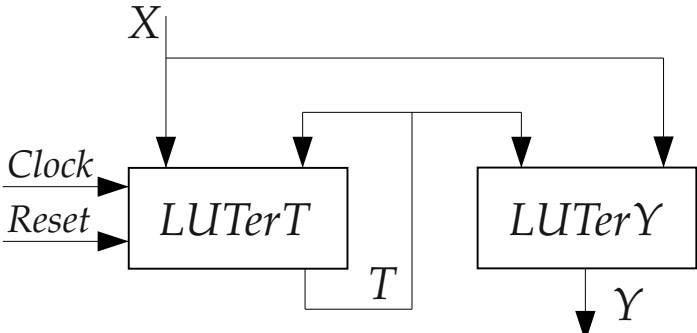

**Figure 1.** Structural diagram of $P$ Mealy FSM.

In $P$ FSMs, the block $LUTerT$ is a block of IMFs. This block implements the SBF (2) and loads the next state code into RG. The register RG is distributed among the LUTs included into CLBs of $LUTerT$. The flip-flops of RG are controlled by pulses $Start$ and $Clock$. The block $LUTerY$ is a block of output logic implementing the SBF (3).

Obviously, the Functions (2) and (3) depend on state variables $T_r \in T$ and FSM inputs $x_l \in X$. Let a function $f_j \in D \cup Y$ depend on $R_j \le R$ state variables and $L_j \le L$ inputs. If the condition

$$R_j + L_j \le S_L\qquad(4)$$

holds, then a corresponding logic circuit consists of a single LUT. If the condition (4) holds for each function $f_j (j \in \{1, \dots, R + N\}$, then the FSM circuit includes exactly $R + N$ LUTs. Such a circuit is single-level. This is the best possible solution providing minimum values of the required chip area, power consumption and cycle time (in other words, the maximum value of operating frequency).

However, FSMs can have up to 10 state variables and 30 inputs [6]. At the same time, the modern LUTs have $S_L = 6$ inputs. So, it is quite possible that condition (4) will be violated for at least a single function $f_j \in D \cup Y$. In this case, it is necessary to use various optimization strategies to optimize the characteristics of an FSM circuit. Our current paper deals with the area reducing problem. Let us analyze some approaches used to solve this problem.

## 3. Relative Works

Methods for solving this problem can be found in a huge number of scientific papers and monographs [10,21,25,27–34]. In the case of LUT-based devices, the occupied chip area is estimated by the required numbers of LUTs (LUT counts) [10]. To diminish the LUT count, three groups of methods are used: (1) the functional decomposition (FD); (2) the optimal state assignment; (3) the structural decomposition (SD). Methods from different groups can be applied simultaneously [30].

In the case of decomposition, Functions (2) and (3) are represented by systems of partial functions [29,35]. Each partial Boolean function has no more than $S_L$ arguments. Due to it, each PBF is represented by a single-LUT circuit. Both FD and SD lead to multi-level FSM circuits. However, these circuits differ in the nature of interconnections [11]. In

the case of FD, the resulting circuit has an irregular interconnect structure in which the same variables $x_l \in X$ and $T_r \in T$ appear at different logical levels of the circuit. In the case of SD, an FSM circuit includes from two to four large logic blocks [30]. These blocks have unique systems of inputs and outputs. Due to it, the SD-based FSM circuits have regular systems of interconnections. As shown in the article [11], SD-based circuits have better characteristics compared to equivalent FD-based circuits. In this article, we discuss a way for improvement some SD-based method.

In the case of LUT-based FSMs, a state assignment is optimal if it allows excluding the maximum possible number of literals from the sum-of-products of Functions (2) and (3) [36]. For the possibility of a single-level implementation of an FSM circuit, it is necessary to exclude such amount of literals that condition (4) is satisfied for each function $f_j \in D \cup Y$. However, this result is possible only for sufficiently simple FSMs [34]. Therefore, in most cases, state encoding methods have an auxiliary nature. If condition (4) is not satisfied after the state assignment, then it is necessary to use other optimization methods.

Very often, the methods of SD are based on finding a partition of the state set $A$ by classes of compatible states. One of such methods is a method of twofold state assignment (TSA) [12,37]. The method is based on construction a partition $\pi_A = \{A^1, \ldots, A^I\}$ of the set A. Each class $A^i \in \pi_A$ determines sets $X^i, Y^i, D^i$. The set $X^i \subseteq X$ includes $L_i$ FSM inputs causing transitions from states $a_m \in A^i$. The set $Y^i \subseteq Y$ consists of FSM outputs produced during the transitions from states $a_m \in A^i$. The set $D^i \subseteq D$ includes input memory functions determining MBCs of transition states.

There are $M_i$ states in each class $A^i \in \pi_A$. Inside each class, these states are encoded by partial maximum binary codes $C(a_m)$ having $R_i$ bits:

$$R_i = \lceil log_2(M_i + 1) \rceil. \tag{5}$$

To encode states $a_m \in A^i$, the variables $v_r \in V^i$ are used. The sets $V^1, \ldots, V^I$ form a set $V$ having $R_A$ elements:

$$R_A = R_1 + \ldots + R_I. \tag{6}$$

A state $a_m \in A$ is compatible with states $a_s \in A^i$, if the including this state into $A^i$ does not violate the following condition:

$$R_i + L_i \leq S_L (i \in \{1, \ldots, I\}). \tag{7}$$

To optimize the FSM logic circuit, it is necessary to minimize the value of $I$. This approach leads to the so-called $P_T$ Mealy FSM (Figure 2).

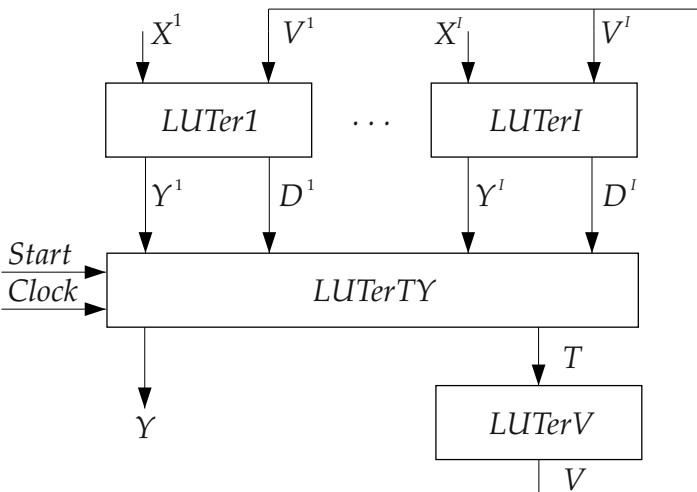

**Figure 2.** Structural diagram of $P_T$ Mealy FSM.

In $P_T$ Mealy FSMs, each state $a_m \in A$ has two codes. These codes are: (1) the maximum binary state code $K(a_m)$ and (2) the partial state code $C(a_m)$ determining a particular state as an element of a particular class. A block *LUTeri* corresponds to the class $A^i \in \pi_A$. This block generates the following systems of PBFs:

$$D^i = D^i(V^i, X^i); \tag{8}$$

$$Y^i = Y^i(V^i, X^i). \tag{9}$$

The LUTerTY creates resulting values of functions $f_j \in D \cup Y$. Each element of LUTerTY implements the following SBFs:

$$D_r = \bigvee_{i=1}^{I} D_r^i (r \in \{1, \ldots, R\}). \tag{10}$$

$$y_n = \bigvee_{i=1}^{I} y_n^i (n \in \{1, \ldots, N\}). \tag{11}$$

The block *LUTerTY* contains the flip-flops of RG. The pulses *Start* and *Clock* enter this block to control the operation of RG.

As follows from (8) and (9), the partial functions depend on state variables $v_r \in V^i$. These state variables are produced by the transformation of the state variables $T_r \in T$. To transform the codes $K(a_m)$, the block *LUTerV* generates the following SBF:

$$V = V(T). \tag{12}$$

As follows from [37], the circuits of $P_T$ FSMs require fewer LUTs than the circuits of equivalent $P$ Mealy FSMs. If the condition

$$I \leq S_L \tag{13}$$

holds, then the circuits of $P_T$ FSMs have exactly three levels of LUTs. As a rule [37], they have higher values of maximum operating frequencies than they are for circuits of equivalent $P$ Mealy FSMs.

We will call the FSM core a block generating partial functions depending on state variables. In $P_T$ FSMs, there is the *CoreV* consisting of blocks *LUTer1-LUTerI*. All other functions are generated by a function assembly block (FAB). In $P_T$ FSMs, the FAB consists of blocks *LUTerTY* and *LUTerV*. Using this terminology, we can represent the structural diagram of $P_T$ FSM in its generalized form (Figure 3).

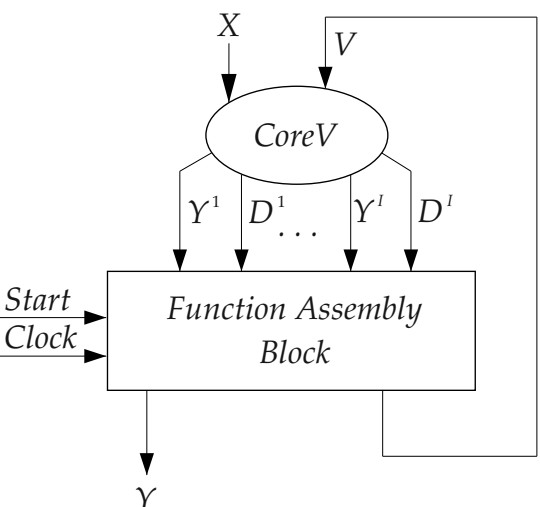

**Figure 3.** Generalized diagram of $P_T$ Mealy FSM.

As follows from Figure 3, all PBFs depend on both inputs $x_l \in X$ and state variables $v_r \in V$. So, the transformation $K(a_m)$ into $C(a_m)$ is executed for all states $a_m \in A$. However, if condition (4) is satisfied for some state $a_m \in A$, then there is no need for the code transformation noted above. If we reduce the number of states whose codes are transformed, then it is possible to reduce both the number of classes ($I$) and the value of the parameter $R_A$. This is an approach proposed in our current paper.

## 4. Main Idea of the Proposed Method

The transitions from a state $a_m \in A$ depend on FSM inputs from a set $X(a_m) \subseteq X$. This set includes $L(a_m) \le L$ elements. Let the following condition hold:

$$L(a_m) + R_{MB} \le S_L. \tag{14}$$

If the condition (14) takes place, then each PBF generated during the transitions from $a_m \in A$ is represented by a single-LUT circuit. So, there is no need in the partial codes for such states $a_m \in A$. So, the partial codes $C(a_m)$ should be generated only for states for which the condition (14) is violated. This conclusion is the basis for a method proposed in this article.

We propose to divide the set $A$ by sets $A_{MB}$ and $A_{PC}$. If the condition (14) holds for a state $a_m \in A$, then this state is included into the set $A_{MB}$. Otherwise, this state is included into the set $A_{PC}$. The states $a_m \in A_{MB}$ form a core denoted as a *CoreT*, whereas the states $a_m \in A_{PC}$ form a core denoted as a *CoreV*. The transformation of state codes is executed only for the states $a_m \in A_{PC}$.

The *CoreT* determines the sets $X_T \subseteq X$, $Y_T \cup Y^0 \subseteq Y$, and $D^0 \subseteq D$. The first set includes FSM inputs determining the transitions from the states $a_m \in A_{MB}$. The second set consists of FSM outputs produced during the transitions from these states. The outputs from the set $Y_T$ are produced only during transitions from the states of the *CoreT*. The outputs from the set $Y^0$ are shared between both cores. The third set includes IMFs generated during the transitions from the states $a_m \in A_{MB}$. The following SBFs determine the *CoreT*:

$$D^0 = D^0(T, X_T); \tag{15}$$

$$Y^0 = Y^0(T, X_T); \tag{16}$$

$$Y_T = Y_T(T, X_T). \tag{17}$$

The *CoreV* determines the sets $X_V \subseteq X$ and $Y_V \subseteq Y$. The first set includes FSM inputs determining the transitions from the states $a_m \in A_{PC}$. The second set consists of FSM outputs produced during the transitions from these states. The following SBFs determine the *CoreT*:

$$D_V^k = D_V^k(V^k, X_V^k); \tag{18}$$

$$Y_V^k = Y_V^k(V^k, X_V^k). \tag{19}$$

The *CoreV* is based on the partition $\pi_V = \{A^1, \ldots, A^K\}$ of the set $A_{PC}$. This partition is constructed in the same way as the partition $\pi_A$. Each class of the partition $\pi_V$ determines the sets $X_V^k$, $Y_V^k$, $V^k$ and $D_V^k$. These sets are similar to the corresponding sets of partial functions considered for the partition $\pi_A$. The circuit of *CoreV* is determined by SBFs similar to SBFs (8) and (9). These SBFs are the following:

$$D = D(T, X_V); \tag{20}$$

$$Y_V = Y_V(T, X_V). \tag{21}$$

To generate the outputs $y_n \in Y_V$ and state variables, it is necessary to use FAB. We propose to combine together the blocks FAB, *CoreV*, and *CoreT*. The proposed connection of blocks leads to a double-core FSM $P_{2C}$. Its generalized structural diagram is shown in Figure 4.

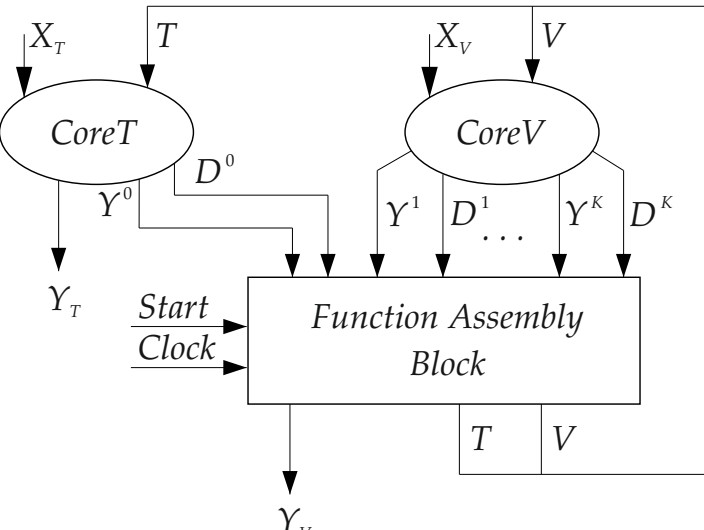

**Figure 4.** Generalized diagram of $P_{2C}$ Mealy FSM.

In Mealy FSM $P_{2C}$, the block *CoreT* implements SBFs (15) and (17). The block *CoreV* generates functions from SBFs (18) and (19). The block FAB includes two blocks, *LUTerTY* and *LUTerV*. The block *LUTerTY* transforms functions (15) and (16), (18) and (19) into resulting values of functions (20) and (21). The block *LUTerV* implements SBF (12).

There are K classes in the partition $\pi_V$. The following condition holds:

$$K \leq I. \tag{22}$$

Then, replacing the subscript $i$ by subscript $k$ turns the Formula (5) into a formula determining the number of state variables in the codes $C(a_m)$ for states $a_m \in A^k$. Having these values allows obtaining the total number of variables $v_r \in V$:

$$R_V = R_1 + R_2 + \ldots + R_K. \tag{23}$$

Obviously, the following condition takes place:

$$R_V \leq R_A. \tag{24}$$

Due to the validity of condition (22), the following is true: (1) the circuit of *CoreV* for FSM $P_{2C}$ must include fewer LUTs than this circuit for the equivalent FSM $P_T$ and (2) the circuit of FSM $P_{2C}$ must include no more levels of logic than it is for the circuit for the equivalent FSM $P_T$. Both $P_T$ and $P_{2C}$ FSMs incorporate the block *LUTerV* executing the transformation of state codes. Obviously, the fewer LUTs has included in the circuit of this block, the less power it consumes. As follows from the validity of condition (24), the circuit of *LUTerV* for FSM $P_{2C}$ must include fewer LUTs than this circuit for the equivalent FSM $P_T$. Therefore, the block *LUTerV* of $P_{2C}$ FSM has less static power consumption than this block of equivalent FSM $P_T$. Since some PBFs are generated by the block *CoreT*, then in some cycles of FSM operation the elements LUTs of the block *LUTerV* do not change their states. So, in these cycles, the block *LUTerV* has the dynamic power consumption close to zero. This analysis suggests that the block *LUTerV* of $P_{2C}$ FSM has less power consumption than that block of an equivalent FSM $P_T$.

So, we assume that the circuits of Mealy FSMs $P_{2C}$ will have fewer LUTs and almost the same or even faster performance compared to circuits of equivalent FSMs $P_T$. We can also argue that $P_{2C}$ FSMs require less energy for the code transformation than equivalent FSMs $P_T$. However, only the experimental studies can show the real energy budgets of equivalent $P_T$ and $P_{2C}$ FSMs.

Using the above information, we propose a method for synthesis of LUT-based $P_{2C}$ Mealy FSMs. As the initial form of FSM representation we use state transition graphs (STGs) [9]. Next, we transform this STG in an equivalent state transition table (STT) [9]. To implement an FSM circuit, we use LUTs having $S_L$ inputs. The proposed method includes the following steps:

1. Transforming the initial STG into STT of *P* Mealy FSM.
2. Preliminary constructing sets $A_{MB}$ and $A_{PC}$.
3. Preliminary constructing the partition $\pi_V$ of the set $A_{PC}$.
4. Redistribution of states between sets $A_{MB}$, $A_{PC}$ and $\pi_V$.
5. Encoding of FSM states by maximum binary codes $K(a_m)$.
6. Creating table of the block *CoreT* and SBFs (15)–(17).
7. Encoding states $a_m \in A^k$ by partial state codes $C(a_m)$.
8. Creating tables of blocks from *CoreV* and SBFs (18) and (19).
9. Creating table of *LUTerTY* and SBFs (20) and (21).
10. Creating table of *LUTerV* and SBF (12).
11. Implementing $P_{2C}$ Mealy FSM circuit using internal resources of a chip.

We use a symbol $P_{2C}(S)$ to show that the model of $P_{2C}$ FSM is used to implement the logic circuit of some FSM *S*. In the next section, we discuss an example of synthesis of $P_{2C}$ Mealy FSM, where we explain how each step is executed.

## 5. Example of Synthesis

We discuss a case of $P_{2C}(S_1)$ FSM synthesis using LUTs with $S_L = 5$. The FSM $S_1$ is represented by an STG shown in Figure 5.

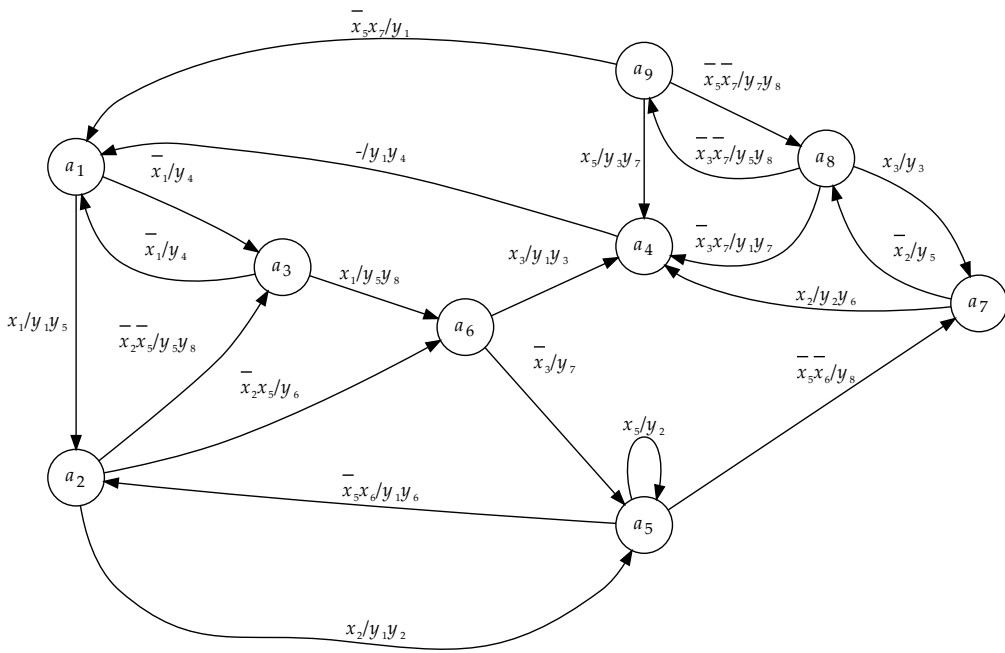

**Figure 5.** State transition graph of Mealy FSM $S_1$.

Each node of an STG corresponds to the FSM state. Each arc of an STG corresponds to an interstate transition [9]. There are *H* arcs in an STG. The *h*-th arc is marked by a pair <input signal $X_h$, collection of outputs $Y_h$>. An input signal $X_h$ is a conjunction of FSM inputs $x_l \in X$ determining the *h*-th interstate transition. A collection of outputs $Y_h \subseteq Y$ includes FSM outputs $y_n \in Y$ generating during the *h*-th interstate transition.

So, the FSM $S_1$ is characterised by the following sets: $A = \{a_1, \ldots, a_9\}$, $X = \{x_1, \ldots, x_7\}$ and $Y = \{y_1, \ldots, y_8\}$. This gives the following values: $M = 9$, $L = 7$, and $N = 8$. As follows from Figure 5, there is $H = 21$.

*Step 1.* This step is executed in the trivial way [6]. Each arc of the STG corresponds to a single line of a corresponding STT. So, this table has the columns $a_m$, $a_s$, $X_h$, $Y_h$, $h$. The state $a_m$ corresponds to a vertex from which the $h$-th arc comes out (this is a current state); the state $a_s$ corresponds to a vertex into which this arc enters (this is a state of transition). The column $X_h$ includes the input signal written above the $h$-th arc. The column $Y_h$ includes the collection of outputs written above the $h$-th arc. Using this approach transforms the STG (Figure 5) into the equivalent STT (Table 1).

**Table 1.** State transition table of Mealy FSM $S_1$.

| $a_m$ | $a_s$ | $X_h$ | $Y_h$ | $h$ |
|---|---|---|---|---|
| $a_1$ | $a_2$ | $x_1$ | $y_1 y_5$ | 1 |
| | $a_3$ | $\overline{x_1}$ | $y_4$ | 2 |
| $a_2$ | $a_5$ | $x_2$ | $y_1 y_2$ | 3 |
| | $a_6$ | $\overline{x_2} x_5$ | $y_6$ | 4 |
| | $a_3$ | $\overline{x_2}\,\overline{x_5}$ | $y_5 y_8$ | 5 |
| $a_3$ | $a_6$ | $x_1$ | $y_5 y_8$ | 6 |
| | $a_1$ | $\overline{x_1}$ | $y_4$ | 7 |
| $a_4$ | $a_1$ | 1 | $y_1 y_4$ | 8 |
| $a_5$ | $a_5$ | $x_5$ | $y_2$ | 9 |
| | $a_2$ | $\overline{x_5} x_6$ | $y_1 y_6$ | 10 |
| | $a_7$ | $\overline{x_5}\,\overline{x_6}$ | $y_8$ | 11 |
| $a_6$ | $a_4$ | $x_3$ | $y_1 y_3$ | 12 |
| | $a_5$ | $\overline{x_3}$ | $y_7$ | 13 |
| $a_7$ | $a_4$ | $x_2$ | $y_2 y_6$ | 14 |
| | $a_8$ | $\overline{x_2}$ | $y_5$ | 15 |
| $a_8$ | $a_7$ | $x_3$ | $y_3$ | 16 |
| | $a_4$ | $\overline{x_3} x_7$ | $y_1 y_7$ | 17 |
| | $a_9$ | $\overline{x_3}\,\overline{x_7}$ | $y_5 y_8$ | 18 |
| $a_9$ | $a_4$ | $x_5$ | $y_3 y_7$ | 19 |
| | $a_1$ | $\overline{x_5} x_7$ | $y_1$ | 20 |
| | $a_8$ | $\overline{x_5}\,\overline{x_7}$ | $y_7 y_8$ | 21 |

*Step 2.* To divide the set $A$ by sets $A_{MB}$ and $A_{PC}$, it is necessary to find values of $L(a_m)$ for states $a_m \in A$. The following values can be found from Table 1: $L(a_4) = 0$; $L(a_m) = 1$ for states $a_1$, $a_3$, $a_6$, $a_7$; $L(a_m) = 2$ for states $a_2$, $a_5$, $a_8$, $a_9$. There is $S_L = 5$. As follows from (14), there are the sets $A_{MB} = \{a_1, a_3, a_4, a_6, a_7\}$ and $A_{PC} = \{a_2, a_5, a_8, a_9\}$. As we show in the next section, some elements of the set $A_{MB}$ can be transferred to the set $A_{PC}$. Thus, these sets do not yet have a final form. Now, we can find sets $X_T$ and $X_V$. The set $X_T$ includes inputs determining transitions from states $a_m \in A_{MB}$, the set $X_V$ includes inputs determining transitions from states $a_m \in A_{PC}$. In the discussed case, there are the following sets: $X_T = \{x_1, x_2, x_3\}$ and $X_V = \{x_2, x_3, x_5, x_6, x_7\}$.

*Step 3.* Using approach [12] gives the partition $\pi_V = \{A^1, A^2\}$ of the set $A_{PC}$. The classes of this partition are the following: $A^1 = \{a_2, a_5\}$ and $A^2 = \{a_8, a_9\}$. This gives the following values of $M_k$: $M_1 = M_2 = 2$. Using (5) gives $R_1 = R_2 = 2$ and $R_V = 4$. Since the set $A_{PC}$ can be changed, the partition $\pi_V$ is also preliminary.

*Step 4.* We discuss this step in Section 6. Now, we only show the outcome of this step. It is the following: $A_{MB} = \{a_1, a_3, a_4\}$ and $A_{PC} = \{a_2, a_5, a_6, a_7, a_8, a_9\}$. Now, the classes of $\pi_V = \{A^1, A^2\}$ are the following: $A^1 = \{a_2, a_5, a_7\}$ and $A^2 = \{a_6, a_8, a_9\}$. This gives the following values of $M_k$ : $M_1 = M_2 = 3$. Using (5) gives $R_1 = R_2 = 2$ and $R_V = 4$. So, there is no change in the total number of state variables $v_r \in V$ before and after refining the sets $A_{MB}$ and $A_{PC}$. So, there is the set $V = \{v_1, \ldots, v_4\}$. However, now there are fewer states in the set $A_{MB}$. This means that the number of LUTs in the circuit of *CoreT* should be reduced compared to this number corresponding to the set $A_{MB}$ obtained during the Step 2.

*Step 5.* There is $M = 9$. Using (1) gives $R_{MB} = 4$. So, there are the following sets: $T = \{T_1, \ldots, T_4\}$ and $D = \{D_1, \ldots, D_4\}$. To minimize the sum-of-products (SOPs) of functions (12), it is necessary to place the states from the same class into minimum possible amount of generalized cubes of $R_{MB}$-dimensional Boolean space [9]. Let us encode the states in a way shown in Figure 6.

**Figure 6.** Outcome of state assignment for Mealy FSM $S_1$.

As follows from Figure 6, the states $a_m \in A_{MB}$ are placed into the cube 00xx. This allows optimizing SOPs of functions (15)–(17). The states $a_m \in A^1$ are placed in the cube x100, the states $a_m \in A^2$ are placed in the cube 1x00. This gives the opportunity to optimize SOPs of functions (12).

*Step 6.* The table of *CoreT* is constructed using the lines 1–2 and 6–8 of Table 1. Three more columns are added in this table: $K(a_m)$, $K(a_s)$ and $D_h^0$. The first and second additional columns include the codes of current and next states, respectively. The column $D_h^0$ includes IMFs equal to 1 to load the code $K(a_s)$ into the RG. We changed the names for columns $X_h$ and $Y_h$ compared to Table 1. Now we use the notation $X_h^0$ and $Y_h^0$. The *CoreT* is represented by Table 2.

**Table 2.** Table of *CoreT* for Mealy FSM $S_1$.

| $a_m$ | $K(a_m)$ | $a_s$ | $K(a_s)$ | $X_h^0$ | $Y_h^0$ | $D_h^0$ | $h$ |
|-------|----------|-------|----------|---------|---------|---------|-----|
| $a_1$ | 0000 | $a_2$ | 0100 | $x_1$ | $y_1 y_5$ | $D_2$ | 1 |
|       |      | $a_3$ | 0001 | $\overline{x_1}$ | $y_4$ | $D_4$ | 2 |
| $a_3$ | 0001 | $a_6$ | 1000 | $x_1$ | $y_5 y_8$ | $D_1$ | 3 |
|       |      | $a_1$ | 0000 | $\overline{x_1}$ | $y_4$ | – | 4 |
| $a_4$ | 0010 | $a_1$ | 0000 | 1 | $y_1 y_4$ | – | 5 |

Using Table 2 gives the following SBFs:

$$
\begin{aligned}
D_1^0 &= F_3^0 = \overline{T_1}\,\overline{T_2}T_4 x_1; & D_2^0 &= F_1^0 = \overline{T_1}\,\overline{T_2}\,\overline{T_3}\,\overline{T_4} x_1; \\
D_4^0 &= F_2^0 = \overline{T_1}\,\overline{T_2}\,\overline{T_3}\,\overline{T_4}\,\overline{x_1}; & y_1^0 &= \overline{T_1}\,\overline{T_2}\,\overline{T_3}\,\overline{T_4} x_1 \vee \overline{T_1}\,\overline{T_2}T_3\,\overline{T_4}; \\
y_4^0 &= [F_2^0 \vee F_4^0] \vee F_5^0 = \overline{T_1}\,\overline{T_2}\,\overline{T_3}\,\overline{x_1} \vee \overline{T_1}\,\overline{T_2}T_3; \\
y_5^0 &= [F_1^0 \vee F_3^0] = \overline{T_1}\,\overline{T_2}\,\overline{T_3} x_1; & y_8^0 &= \overline{T_1}\,\overline{T_2}T_4 x_1.
\end{aligned}
\tag{25}
$$

This system is used to create the circuit of *CoreT*. Let us point out that the function $y_4$ is generated only by some LUT of *CoreT*. This gives $Y_T = \{y_4\}$. Furthermore, the following sets can be derived from Table 2: $X^0 = \{x_1\}$, $Y^0 = \{y_1, y_5, y_8\}$ and $D^0 = \{D_1, D_2, D_4\}$.

*Step 7.* To encode the states $a_m \in A^1$, the variables $v_1, v_2 \in V$ are used. To encode the states $a_m \in A^2$, the variables $v_3, v_4 \in V$ are used. We use the code 00xx to show that a particular state does not belong to the class $A^1$. The code xx00 shows that a particular state does not belong to the class $A^2$. The outcome of state assignment is shown in Figure 7.

**Figure 7.** Partial state codes for Mealy FSM $S_1$.

The following partial codes can be found from the Karnaugh map (Figure 7): $C(a_2) = C(a_6) = 01$, $C(a_5) = C(a_8) = 10$, and $C(a_7) = C(a_9) = 11$. These codes are used in LUTs of *CoreV*.

*Step 8.* There are two blocks of LUTs in the *CoreV*. The block *LUTer1* implements SBFs for the class $A^1$; the block *LUTer2* implements SBFs for the class $A^2$. The table of *LUTer1* is constructed using the lines 3–5, 9–11 and 14–15 of Table 1. This is Table 3. The table of *LUTer2* is constructed using the lines 12–13 and 16–21 of Table 1. This is Table 4.

**Table 3.** Table of *LUTer1*.

| $a_m$ | $C(a_m)$ | $a_s$ | $K(a_s)$ | $X_h^1$ | $Y_h^1$ | $D_h^1$ | $h$ |
|-------|----------|-------|----------|---------|---------|---------|-----|
| $a_2$ | 01 | $a_5$ | 0101 | $x_2$ | $y_1 y_2$ | $D_2 D_4$ | 1 |
|       |          | $a_6$ | 1000 | $\overline{x_2} x_5$ | $y_6$ | $D_1$ | 2 |
|       |          | $a_3$ | 0001 | $\overline{x_2}\ \overline{x_5}$ | $y_5 y_8$ | $D_4$ | 3 |
| $a_5$ | 10 | $a_5$ | 0101 | $x_5$ | $y_2$ | $D_2 D_4$ | 4 |
|       |          | $a_2$ | 0100 | $\overline{x_5} x_6$ | $y_1 y_6$ | $D_2$ | 5 |
|       |          | $a_7$ | 0110 | $\overline{x_5}\ \overline{x_6}$ | $y_8$ | $D_2 D_3$ | 6 |
| $a_7$ | 11 | $a_4$ | 0010 | $x_2$ | $y_2 y_6$ | $D_3$ | 7 |
|       |          | $a_8$ | 1001 | $\overline{x_2}$ | $y_5$ | $D_1 D_4$ | 8 |

**Table 4.** Table of *LUTer2*.

| $a_m$ | $C(a_m)$ | $a_s$ | $K(a_s)$ | $X_h^2$ | $Y_h^2$ | $D_h^2$ | $h$ |
|-------|----------|-------|----------|---------|---------|---------|-----|
| $a_6$ | 01 | $a_4$ | 0010 | $x_3$ | $y_1 y_3$ | $D_3$ | 1 |
|       |          | $a_5$ | 0101 | $\overline{x_3}$ | $y_7$ | $D_2 D_4$ | 2 |
| $a_8$ | 10 | $a_7$ | 0110 | $x_3$ | $y_3$ | $D_2 D_3$ | 3 |
|       |          | $a_4$ | 0010 | $\overline{x_3} x_7$ | $y_1 y_7$ | $D_3$ | 4 |
|       |          | $a_9$ | 1010 | $\overline{x_3}\ \overline{x_7}$ | $y_5 y_8$ | $D_1 D_3$ | 5 |
| $a_9$ | 11 | $a_4$ | 0010 | $x_5$ | $y_3 y_7$ | $D_2$ | 6 |
|       |          | $a_1$ | 0000 | $\overline{x_5} x_7$ | $y_1$ | – | 7 |
|       |          | $a_8$ | 1001 | $\overline{x_5}\ \overline{x_7}$ | $y_7 y_8$ | $D_1 D_4$ | 8 |

Both tables use partial state codes $C(a_m)$ for current states and the MBCs $K(a_s)$ for states of transition. The following sets can be found from Table 3: $X^1 = \{x_2, x_5, x_6\}$, $Y^1 = \{y_1, y_2, y_5, y_6, y_8\}$ and $D^1 = D$. The following sets can be found from Table 4: $X^2 = \{x_3, x_5, x_7\}$, $Y^2 = \{y_1, y_3, y_5, y_7, y_8\}$ and $D^2 = D$.

The SBFs (18) and (19) are constructed in the same way as this is for SBFs (15)–(17). For example, the following SOPs can be obtained for functions $D_1^1$ (Table 3) and $D_1^2$ (Table 4):

$$D_1^1 = F_2^1 \vee F_8^1 = \overline{v_1} v_2 \overline{x_2} x_5 \vee v_1 v_2 \overline{x_2};$$
$$D_1^2 = F_5^2 \vee F_8^2 = v_3 \overline{v_4}\ \overline{x_3}\ \overline{x_7} \vee v_3 v_4 \overline{x_5}\ \overline{x_7}. \tag{26}$$

*Step 9.* There are the following columns in table of *LUTerTY*: $f_j$ (a function generated by $\overline{LUTerTY}$), *CoreT*, *CoreV*. If a function $f_j \in D \in Y$ is generated by a LUT of *CoreT*, then there is 1 in the intersection of the line with this function and the column of the corresponding core. Otherwise, this intersection is marked by 0. There are $K$ sub-columns in the column *CoreV*. If a function $f_j \in D \cup Y$ is generated by $LUTerk$ of *CoreV*, then there is 1 in the intersection of the line with this function and the sub-column k. In the discussed case, the block *LUTerTY* is represented by Table 5.

**Table 5.** Table of *LUTerTY*.

| $f_j$ | *CoreT* | *CoreV* | |
|---|---|---|---|
| | | 1 | 2 |
| $D_1$ | 1 | 1 | 1 |
| $D_2$ | 1 | 1 | 1 |
| $D_3$ | 0 | 1 | 1 |
| $D_4$ | 1 | 1 | 1 |
| $y_1$ | 1 | 1 | 1 |
| $y_2$ | 0 | 1 | 0 |
| $y_3$ | 0 | 0 | 1 |
| $y_4$ | 1 | 0 | 0 |
| $y_5$ | 1 | 1 | 1 |
| $y_6$ | 0 | 1 | 0 |
| $y_7$ | 0 | 0 | 1 |
| $y_8$ | 1 | 1 | 1 |

To fill the column *CoreT*, the data from Table 2 are used. To fill the sub-column 1, we use Table 3. Table 4 is a base for filling the sub-column 2. We hope there is a transparent connection between Tables 2–5.

Using Table 5, we can construct the following SBFs:

$$
\begin{aligned}
D_1 &= D_1^0 \vee D_1^1 \vee D_1^2; & D_2 &= D_2^0 \vee D_2^1 \vee D_2^2; \\
D_3 &= D_3^1 \vee D_3^2; & D_4 &= D_4^0 \vee D_4^1 \vee D_4^2; \\
y_1 &= y_1^0 \vee y_1^1 \vee y_1^2 & y_2 &= y_2^1; & y_3 &= y_3^2; \\
y_4 &= y_4^0; & y_5 &= y_5^0 \vee y_5^1 \vee y_5^2; & y_6 &= y_6^1; \\
y_7 &= y_7^2; & y_8 &= y_8^0 \vee y_8^1 \vee y_8^2.
\end{aligned}
\tag{27}
$$

Each function $f_j \in D \cup Y$ is represented by a disjunction of its partial components. The principle of constructing each function of (27) is clear from the comparison of these functions with contents of Table 5.

*Step 10.* To create the table of *LUTerV*, we should use the full codes $K(a_m)$ and partial state codes $C(a_m)$. So, there are the following columns in this table: $a_m$, $K(a_m)$, $C(a_m)$, $V_m$. Inside this table, we use only states $a_m \in A_{PC}$. In the discussed case, there are six lines in the table of *LUTerV* (Table 6).

**Table 6.** Table of *LUTerV*.

| $a_m$ | $K(a_m)$ | $C(a_m)$ | $V_m$ |
|---|---|---|---|
| $a_2$ | 0100 | 0100 | $v_2$ |
| $a_5$ | 0101 | 1000 | $v_1$ |
| $a_6$ | 1000 | 0001 | $v_4$ |
| $a_7$ | 0110 | 1100 | $v_1 v_2$ |
| $a_8$ | 1001 | 0010 | $v_3$ |
| $a_9$ | 1010 | 0011 | $v_3 v_4$ |

To fill the column $K(a_m)$, we use the state codes from Figure 6. The column $C(a_m)$ is filled using the partial state codes from Figure 7.

To optimize the SBF (12), we represent its functions by the Karnaugh map (Figure 8). In this map, we treat the codes of states $a_m \in A_{MB}$ as the "don't care" input assignment.

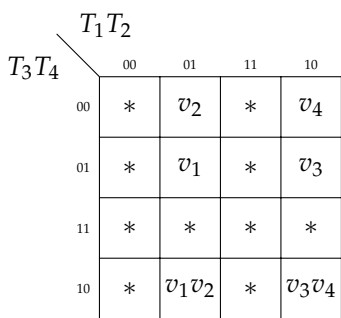

**Figure 8.** Karnaugh map for SBF $V(T)$.

Using the Karnaugh map (Figure 8) gives the following SBF:

$$
\begin{aligned}
v_1 &= \overline{T_1}T_4 \vee \overline{T_1}T_3; \\
v_2 &= \overline{T_1}\,\overline{T_4}; \\
v_3 &= T_1 T_4 \vee T_1 T_3; \\
v_4 &= T_1\overline{T_4}.
\end{aligned}
\tag{28}
$$

In the worst case, each function $v_r \in V$ is represented by a SOP having $R_{MB}$ literals. So, the maximum number of literals is calculated as the product of $R_V$ by $R_{MB}$. In the discussed case, this number is equal to 16. If we analyze the SBF (28), we find that it includes 10 literals. So, using our approach allows reducing the number of literals by a factor of 1.6. Each literal corresponds to an interconnection between outputs of RG and inputs of LUTs creating the circuit of *LUTerV*. It is known that minimizing the number of interconnections allows reducing the value of power consumption [26,38].

*Step 11.* To implement the circuit of $P_{2C}$ Mealy FSM, it is necessary to use, for example, the CAD tool Vivado by Xilinx [39]. This package solves all problems connected with the step of technology mapping [40,41]. In Section 7, we use Vivado to compare the proposed method with some known FSM design methods.

## 6. Algorithm of State Redistribution

If a class $A^k \in \pi_V$ includes $M_k$ states, then it is necessary $R_k$ state variables to encode the states $a_m \in A^k$ by the partial state codes $C(a_m)$. The value of $R_k$ is determined by (5). We denote as $MNP_k$ the maximum possible number of states in a class $A^k \in \pi_V$. This value is determined as

$$
MNP_k = 2^{R_k} - 1.
\tag{29}
$$

Our research shows that it is quite possible that some class $A^k \in \pi_V$ includes fewer states compared to the value of $MNP_k$. For example, we have the following classes for FSM $S_1$: $A^1 = \{a_2, a_5\}$ and $A^2 = \{a_8, a_9\}$. Using (5) gives $R_1 = R_2 = 2$. Using (29) gives $MNP_1 = MNP_2 = 3$. So, both classes might be supplemented by states from the set $A_{MB} = \{a_1, a_3, a_4, a_6, a_7\}$. One state can be added to each of the classes $A^k \in \pi_V$.

So, it is quite possible that we need to redistribute states between sets $A_{MB}$ and $A_{PC}$. Obviously, these new elements of $A_{PC}$ should be added into some classes $A^k \in \pi_V$. It is obvious that it is expedient to transfer states in such a way as to reduce the number of states in the set $A_{MB}$ as much as possible.

We propose to use an estimate $I(a_m)$, which we called the influence of the state $a_m \in A_{MB}$ on the sets $X_T$ and $X_V$. In the discussed case, these sets are the following: $X_T = \{x_1, x_2, x_3\}$ and $X_V = \{x_2, x_3, x_5, x_6, x_7\}$.

The best candidate for transfer to the set $A^k \in \pi_V$ is the state $a_m \in A_{MB}$ that minimizes the number of inputs in the set $X_T$ and minimally increases this number in the set $X^k$. The influence of a state $a_m \in A_{MB}$ on the set $X_T$ is determined as

$$
I_T(a_m) = |X(a_m) \backslash X_T|.
\tag{30}
$$

The influence of a state $a_m \in A_{MB}$ on the set $X^k$ is determined as

$$I_V(a_m) = |X(a_m) \backslash X^k|. \tag{31}$$

So, the overall influence of the state $a_m \in A_{MB}$ is defined as

$$I(a_m) = I_T(a_m) - I_V(a_m). \tag{32}$$

Obviously, it is necessary to transfer the states with the greatest influence. This is the basis of our proposed redistribution algorithm (Figure 9).

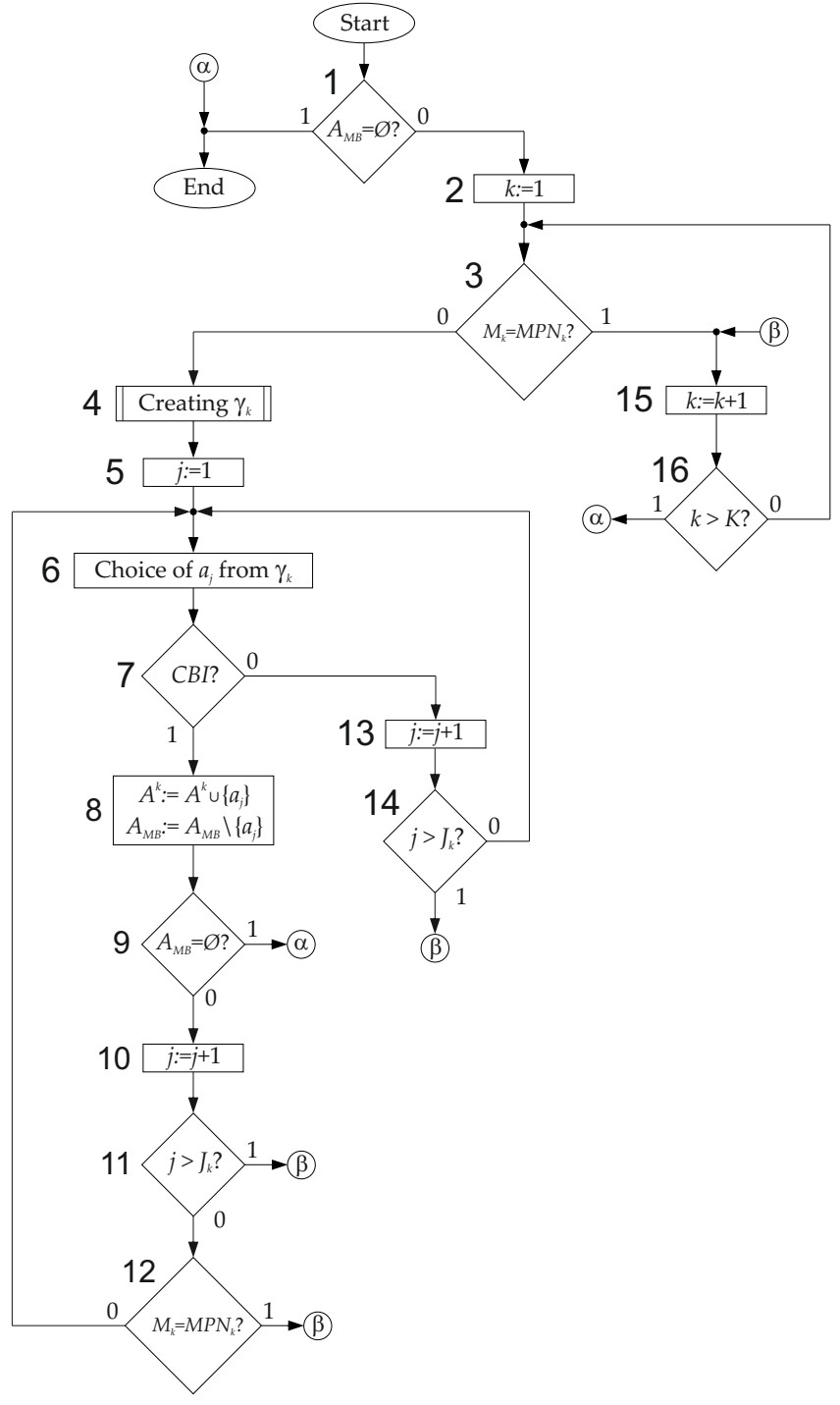

**Figure 9.** Block-diagram of state redistribution algorithm.

During the redistribution, a queue $\gamma_k$ is formed from the states $a_m \in A_{MB}$. This queue is based on the following rule: the states are placed as the value of $I(a_m)$ decreases. If the influence is the same for states $a_m, a_s \in A_{MB}(I(a_m) = I(a_s))$, then, in the queue, the state with lower subscript precedes a state with higher subscript. A state can be included into a class $A^k \in \pi_V$, if its including does not violate the condition (4). In our algorithm, we use the abbreviation CBI (can be included). For each class $A^k \in \pi_V$, the queue $\gamma_k$ includes $J_k$ elements. This preliminary information is quite enough to proceed to the description of the proposed algorithm.

We start the redistribution from the testing the set $A_{MB}$ (Block 1). If this set is empty (output 1), then the redistribution cannot be executed. If there are some states in the set $A_{MB}$ (output 0), then the redistribution process begins. The analysis starts with class $A^1 \in \pi_V$ (Block 2). If the analyzed class includes the maximum number of states (output 1 from Block 3), then it is necessary to proceed to the analysis of the next class (go to Block 15). The algorithm is terminated when all classes are analyzed (output 1 of Block 16). Otherwise, the next class is analyzed (go to from Block 16 to Block 3).

If an additional state can be included in the class $A^k \in \pi_V$ (output 0 from Block 3), then there is created a queue $\gamma_k$ having $J_k$ elements (Block 4). Next, the sequential analysis of the states from the queue $\gamma_k$ is performed. The analysis starts from the first element of the queue (Block 5).

The $j$-th element is taken from the queue (Block 6). If it cannot be included into the class $A^k \in \pi_V$ (output 0 from Block 7), then the next element of the queue should be analyzed (go to Block 13). If all elements are analyzed (output 1 of Block 14), then it is necessary to analyze the class $A^{k+1} \in \pi_V$ (go to Block 15). Otherwise (output 0 of Block 14), the next element of the queue is analyzed (go to Block 6).

If the $j$-th element can be included into the class $A^k \in \pi_V$ (output 1 from Block 7), then the following actions are executed (Block 8): (1) the state $a_j \in A_{MB}$ is included into the set $A^k \in \pi_V$; (2) the state $a_j \in A_{MB}$ is excluded from the set $A_{MB}$. If now (after excluding state $a_j \in A_{MB}$) the set $A_{MB}$ becomes empty (output 1 of Block 9), the redistribution process is terminated (go to End). Otherwise (output 0 of Block 9), the next element of queue should be analyzed (go to Block 10). If all elements are already analyzed (output 1 of Block 11), then it is necessary to analyze the class $A^{k+1} \in \pi_V$ (go to Block 15). Otherwise (output 0 of Block 11), the next element of queue should be analyzed. This can be done if the class $A^k \in \pi_V$ does not contain the maximum possible number of elements. This is checked in the Block 12. If the class is full (output 1 of Block 12), then it is necessary to analyze the class $A^{k+1} \in \pi_V$ (go to Block 15). Otherwise (output 0 of Block 12), the next element of the queue is analyzed (go to Block 6).

There are two conditions to terminate this redistribution process. First, if there are no elements in the set $A_{MB}$ (outputs 1 from Blocks 1 and 9). Second, all classes $A^k \in \pi_V$ have been tested and, if it was possible, supplemented by states $a_m \in A_{MB}$ (output 1 from Block 16).

So, the $k$-th step of the redistribution process starts from creating current sets $A_{MB}$ and $X^0$. Next, it is necessary to find the values of $I(a_m)$ for states $a_m \in A_{MB}$ and create the current queue $\gamma_k$. So, there are $K$ columns corresponding to classes $A^k \in \pi_V$ in the table of redistribution. Each column is divided by the following sub-columns: $A_{MB}$, $I(a_m)$, $\gamma_k$, $j = 1$, $j = 2, \ldots, j = J_k$. In this table, the line $a_m$ includes states $a_m \in A_{MB}$ transferred in the particular class $A^k \in \pi_V$. The lines for these states are marked by $\oplus$. If a state cannot be included into the class $A^k \in \pi_V$, the corresponding line includes the sign "$-$". The last line of the table contains the classes $A^k \in \pi_V$. Table 7 shows the redistribution process for FSM $S_1$.

**Table 7.** Redistribution process for FSM $S_1$.

| | | $k = 1$ | | | | | $k = 2$ | | |
|---|---|---|---|---|---|---|---|---|---|
| $A_{MB}$ | $I(a_m)$ | $\gamma_1$ | $j = 1$ | $j = 2$ | $A_{MB}$ | $I(a_m)$ | $\gamma_2$ | $j = 1$ | $j = 2$ |
| $a_1$ | $-1$ | 4 | | $-$ | $a_1$ | $-1$ | 3 | | $-$ |
| $a_3$ | $-1$ | 5 | | $-$ | $a_3$ | $-1$ | 4 | | $-$ |
| $a_4$ | 0 | 2 | | $-$ | $a_4$ | 0 | 2 | | $-$ |
| $a_6$ | 0 | 3 | | $-$ | $a_6$ | 1 | 1 | $\oplus$ | $-$ |
| $a_7$ | 1 | 1 | $\oplus$ | $-$ | $-$ | $-$ | $-$ | $-$ | $-$ |
| $a_m$ | | $a_7$ | | | $a_m$ | | $a_6$ | | |
| $A^1$ | | $a_2 a_5 a_7$ | | | $A^2$ | | $a_8 a_9 a_6$ | | |

Let us go back to the previous section. After executing the step 2, we have the following sets: $A_{MB} = \{a_1, a_3, a_4, a_6, a_7\}$, $A_{PC} = \{a_2, a_5, a_8, a_9\}$, $X_T = \{x_1, x_2, x_3\}$ and $X_V = \{x_2, x_3, x_5, x_6, x_7\}$. After executing the step 3, we have the partition $\pi_V = \{A^1, A^2\}$ with the following classes: $A^1 = \{a_2, a_5\}$ and $A^2 = \{a_8, a_9\}$. These classes are characterized by the sets $X^1 = \{x_2, x_5, x_6\}$ and $X^2 = \{x_3, x_5, x_7\}$.

So, for $k = 1$, the column $A_{MB}$ contains the states $a_1, a_3, a_4, a_6, a_7$. For the state $a_1 \in A_{MB}$, we can find the set $X(a_1) = \{x_1\}$. Let us find the value of $I(a_1)$. Using (30) gives the following: $I_T(a_1) = |X(a_1) \backslash X_T| = |\{x_1\} \backslash \{x_1, x_2, x_3\}| = 0$. Using (31) gives $I_V(a_1) = |X(a_1) \backslash X^1| = |\{x_1\} \backslash \{x_2, x_5, x_6\}| = 1$. Using (32) gives $I(a_1) = I_T(a_1) - I_V(a_1) = 0 - 1 = -1$. This value is written in the intersection of the line $a_1$ and sub-column $I(a_m)$ for $k = 1$. In the same way, the values of $I(a_m)$ for all other states $a_m \in A_{MB}$ are calculated.

Using the values of $I(a_m)$, we can get the queue $\gamma_1 = < a_7, a_4, a_6, a_1, a_3 >$. In the intersection of the line $a_m$ and the sub-column $\gamma_1$, there is written the place of this state in this queue. So, we should check the possibility of redistribution starting from the state $a_7$. If we place the state $a_7$ into the class $A^1$, then there is no change for values of $L_1$ and $R_1$. So, the state is included into $A^1$ and excluded from $A_{MB}$. Now, there is $M_1 = MPN_1 = 3$. So, during the step $j = 2$ no state can be added into the class $A^1$.

Now, there are the following modified sets: $A^1 = \{a_2, a_5, a_7\}$, $A_{MB} = \{a_1, a_3, a_4, a_6\}$ and $X_T = \{x_1, x_3\}$. Using the modified sets $A_{MB}$ and $X_T$, we can start the next step of redistribution ($k = 2$).

The values of $I(a_m)$ are shown in the corresponding sub-column of the column $k = 2$. Using them gives the queue $\gamma_2 = < a_6, a_4, a_1, a_3 >$. If we place the state $a_6$ into the class $A^2$, then there is no change for values of $L_2$ and $R_2$. So, the state $a_6$ is included into $A^2$ and excluded from $A_{MB}$. Now, there is $M_2 = MPN_2 = 3$. So, during the step $j = 2$ no state can be added into the class $A^2$. So, the class $A^2$ is ready.

Now, there are the following modified sets: $A^1 = \{a_2, a_5, a_7\}$, $A^2 = \{a_6, a_8, a_9\}$, $A_{MB} = \{a_1, a_3, a_4\}$ and $X_T = \{x_1\}$. Obviously, these sets are the same as we use as the outcome of Step 4 in our example.

## 7. Experimental Results

In this section, the results of experiments conducted with the benchmarks [42] are shown. The library [42] consists of 48 benchmarks. The benchmark FSMs are represented by their STTs. To represent the STTs, the format KISS2 is used. These benchmarks have a wide range of basic characteristics (numbers of states, inputs, and outputs). Different researchers use these benchmarks to compare various characteristics of FSM circuits [28,29,32]. The characteristics of benchmarks are shown in Table 8.

Our current research is connected with Mealy FSMs which are the parts of digital systems. It is known that Mealy FSMs are not stable [6],: fluctuations at the inputs lead to fluctuations at the outputs. This can lead to errors in the operation of the digital system as a whole. To avoid these errors, the FSM inputs should be stabilized. The stabilization presumes using an additional input register (AIR) [30]. When input values stabilize, they are loaded into the AIR. Now, fluctuations at the inputs (which are the outputs of some system's blocks) do not lead to fluctuations at the FSM outputs. However, the AIR consumes some resources of a chip: (1) it requires L additional LUTs and flip-flops and

(2) it is synchronized (due to it, AIR uses some resources of the synchronization tree). So, this register consumes additional LUTs, flip-flops, power and time (it adds some delay to the whole synchronization cycle time). Such an approach allows taking into account this overhead connected with the stabilization of FSM operation.

**Table 8.** Characteristics of benchmarks [42].

| Benchmark | $L$ | $N$ | $R_{MB} + L$ | $M/R_{MB}$ | $H$ | Class |
|-----------|-----|-----|--------------|------------|-----|-------|
| bbara | 4 | 2 | 8 | 12/4 | 60 | 1 |
| bbsse | 7 | 7 | 12 | 26/5 | 56 | 1 |
| bbtas | 2 | 2 | 6 | 9/4 | 24 | 0 |
| beecount | 3 | 4 | 7 | 10/4 | 28 | 1 |
| cse | 7 | 7 | 12 | 32/5 | 91 | 1 |
| dk14 | 3 | 5 | 8 | 26/5 | 56 | 1 |
| dk15 | 3 | 5 | 8 | 17/5 | 32 | 1 |
| dk16 | 2 | 3 | 9 | 75/7 | 108 | 1 |
| dk17 | 2 | 3 | 6 | 16/4 | 32 | 0 |
| dk27 | 1 | 2 | 5 | 10/4 | 14 | 0 |
| dk512 | 1 | 3 | 6 | 24/5 | 15 | 0 |
| donfile | 2 | 1 | 7 | 24/5 | 96 | 1 |
| ex1 | 9 | 19 | 16 | 80/7 | 138 | 2 |
| ex2 | 2 | 2 | 7 | 25/5 | 72 | 1 |
| ex3 | 2 | 2 | 6 | 14/4 | 36 | 0 |
| ex4 | 6 | 9 | 11 | 18/5 | 21 | 1 |
| ex5 | 2 | 2 | 6 | 16/4 | 32 | 0 |
| ex6 | 5 | 8 | 9 | 14/4 | 34 | 1 |
| ex7 | 2 | 2 | 12 | 17/5 | 36 | 1 |
| keyb | 7 | 7 | 12 | 22/5 | 170 | 1 |
| kirkman | 12 | 6 | 18 | 48/6 | 370 | 2 |
| lion | 2 | 1 | 5 | 5/3 | 11 | 0 |
| lion9 | 2 | 1 | 6 | 11/4 | 25 | 0 |
| mark1 | 5 | 16 | 10 | 22/5 | 22 | 1 |
| mc | 3 | 5 | 6 | 8/3 | 10 | 0 |
| modulo12 | 1 | 1 | 5 | 12/4 | 24 | 0 |
| opus | 5 | 6 | 10 | 18/5 | 22 | 1 |
| planet | 7 | 19 | 14 | 86/7 | 115 | 2 |
| planet1 | 7 | 19 | 14 | 86/7 | 115 | 2 |
| pma | 8 | 8 | 14 | 49/6 | 73 | 2 |
| s1 | 8 | 7 | 14 | 54/6 | 106 | 2 |
| s1488 | 8 | 19 | 15 | 112/7 | 251 | 2 |
| s1494 | 8 | 19 | 15 | 118/7 | 250 | 2 |
| s1a | 8 | 6 | 15 | 86/7 | 107 | 2 |
| s208 | 11 | 2 | 17 | 37/6 | 153 | 2 |
| s27 | 4 | 1 | 8 | 11/4 | 34 | 1 |
| s386 | 7 | 7 | 12 | 23/5 | 64 | 1 |
| s420 | 19 | 2 | 27 | 137/8 | 137 | 4 |
| s510 | 19 | 7 | 27 | 172/8 | 77 | 4 |
| s8 | 4 | 1 | 8 | 15/4 | 20 | 1 |
| s820 | 18 | 19 | 25 | 78/7 | 232 | 4 |
| s832 | 18 | 19 | 25 | 76/7 | 245 | 4 |
| sand | 11 | 9 | 18 | 88/7 | 184 | 3 |
| shiftreg | 1 | 1 | 5 | 16/4 | 16 | 0 |
| sse | 7 | 7 | 12 | 26/5 | 56 | 1 |
| styr | 9 | 10 | 16 | 67/7 | 166 | 2 |
| tma | 7 | 9 | 13 | 63/6 | 44 | 2 |

The experiments are conducted using a personal computer with the following characteristics: CPU: Intel Core i5-11300H, Memory: 16GB RAM LPDDR4X. To get the FSM circuits, we use the Virtex-7 VC709 Evaluation Platform (xc7vx690tffg1761-2) [43] by AMD Xilinx. There is $S_L = 6$ for LUTs used in this platform includes. The CAD tool Vivado

v2019.1 (64-bit) [39] executes the technology mapping. The results of experiments are taken from reports produced by Vivado. To connect the library with Vivado, we use VHDL-based FSM models. These models are obtained by a transformation of the files in KISS2 format into VHDL codes. The transformation is executed by the CAD tool K2F [30].

We have found three main characteristics of $P_{2C}$ Mealy FSMs. They are: the occupied chip area (the LUT count), performance (both the values of cycle time and maximum operating frequency), and power consumption. We compared the obtained values with the corresponding values for four different FSMs. Three of them are $P$ Mealy FSMs based on: (1) Auto of Vivado (it uses MBCs); (2) One-hot of Vivado; (3) JEDI (it uses MBCs, too). Moreover, for the comparison, we use $P_T$-based FSMs [12] whose circuits we try to improve.

As shown in [30], all main characteristics of LUT-based FSM circuits depend on the relation between the values of $L + R_{MB}$, on the one hand, and the value of $S_L$, on the other hand:

$$nS_L < (L + R_{MB}) \leq (n + 1)S_L. \tag{33}$$

Analysis of Table 8 allows dividing the benchmarks into five sets. The benchmarks belong to class of trivial FSMs (set 0), if $n = 0$ (it gives $R_{MB} + L \leq 6$). I The benchmarks belong to set of simple FSMs (set 1), if $n = 1$ (it gives $R_{MB} + L \leq 12$). The benchmarks belong to set of average FSMs (set 2), if $n = 2$ (it gives $R_{MB} + L \leq 18$). The benchmarks belong to set of big FSMs (set 3), if $n = 3$ (it gives $R_{MB} + L \leq 24$). The benchmarks belong to set of very big FSMs (set 4), if $n = 4$ (it gives the relation $R_{MB} + L > 24$). As research [37] shows, the larger the set number, the bigger the gain from using methods of twofold state assignment.

The results of experiments are shown in Tables 9–11. These tables are organized in the same manner. The table columns are marked by the names of investigated methods. The last column includes the number of the benchmark set to whom the particular benchmark belongs. The table rows are marked the names of benchmarks. There are results of summation of values from columns in the row "Total". The row "Percentage" includes the percentage of summarized characteristics of FSM circuits produced by other methods respectively to $P_{2C}$-based FSMs. We start the analysis of experiments from Table 9. This table contains the values of LUT counts for each benchmark used in the experiments.

As follows from Table 9, the circuits of $P_{2C}$-based FSMs use a minimum number of LUTs compared to other investigated methods. There is the following gain: (1) 36.92% compared to Auto-based FSMs; (2) 56.23% compared to One-hot–based FSMs; (3) 16.11% compared to JEDI-based FSMs; (4) 5.74% compared to $P_T$-based FSMs. In our opinion, this gain is associated with a decrease in the number of variables used in partial state codes (compared to equivalent $P_T$-based FSMs). The second source of a decrease in the LUT counts can be a decrease in the number of partition classes. If the relation $(K + 1) < I$ takes place, then there is a decrease in the required number of LUT inputs for elements of $LUTerTY$. If the condition (13) is violated but the condition $(K + 1) < S_L$ holds, then the circuit of $LUTerTY$ is multi-level for a $P_T$-based FSM as opposed to the single-level block circuit of an equivalent $P_{2C}$-based FSM.

Careful analysis of the table reveals the following feature of the proposed method: there are the same values of LUT counts for equivalent $P_T$- and $P_{2C}$-based FSMs for the Set 0. This can be explained as follows. For this set, the condition (14) holds. This means that each function $f_j \in D \cup Y$ does not require being decomposed. Only a single LUT is enough to implement a logic circuit for any function $f_j \in D \cup Y$. In this case, there is the same single class into both partitions, $\pi_A$ and $\pi_V$. Due to it, the block FAB is absent. This means that both $P_T$ and $P_{2C}$ FSMs turn into $P$ FSMs. So, there are the same circuits for $P_T$ and $P_{2C}$ FSMs. Obviously, these circuits have the same values of LUT counts. The same should take place also for other characteristics of these two models.

**Table 9.** Experimental results (the LUT counts).

| Benchmark | Auto | One-Hot | JEDI | $P_T$ | Our Approach | Set |
|---|---|---|---|---|---|---|
| bbara | 21 | 21 | 14 | 15 | 14 | 1 |
| bbsse | 40 | 44 | 31 | 29 | 24 | 1 |
| bbtas | 7 | 7 | 7 | 7 | 7 | 0 |
| beecount | 22 | 22 | 17 | 15 | 13 | 1 |
| cse | 47 | 73 | 43 | 39 | 36 | 1 |
| dk14 | 19 | 30 | 13 | 15 | 13 | 1 |
| dk15 | 18 | 19 | 15 | 9 | 9 | 1 |
| dk16 | 17 | 36 | 14 | 12 | 12 | 1 |
| dk17 | 7 | 14 | 7 | 7 | 7 | 0 |
| dk27 | 4 | 6 | 5 | 7 | 7 | 0 |
| dk512 | 11 | 11 | 10 | 9 | 9 | 0 |
| donfile | 33 | 33 | 26 | 21 | 18 | 1 |
| ex1 | 79 | 83 | 62 | 51 | 51 | 2 |
| ex2 | 11 | 11 | 10 | 10 | 9 | 1 |
| ex3 | 11 | 11 | 11 | 10 | 9 | 0 |
| ex4 | 21 | 19 | 18 | 16 | 13 | 1 |
| ex5 | 11 | 11 | 11 | 10 | 10 | 0 |
| ex6 | 29 | 41 | 27 | 25 | 20 | 1 |
| ex7 | 6 | 7 | 6 | 6 | 6 | 1 |
| keyb | 50 | 68 | 47 | 44 | 39 | 1 |
| kirkman | 54 | 70 | 51 | 47 | 39 | 2 |
| lion | 4 | 7 | 4 | 4 | 4 | 0 |
| lion9 | 8 | 13 | 7 | 7 | 7 | 0 |
| mark1 | 28 | 28 | 25 | 23 | 20 | 1 |
| mc | 7 | 10 | 7 | 7 | 7 | 0 |
| modulo12 | 8 | 8 | 8 | 8 | 8 | 0 |
| opus | 33 | 33 | 27 | 28 | 24 | 1 |
| planet | 138 | 138 | 95 | 87 | 87 | 2 |
| planet1 | 138 | 138 | 95 | 87 | 87 | 2 |
| pma | 102 | 102 | 94 | 86 | 77 | 2 |
| s1 | 73 | 107 | 69 | 65 | 59 | 2 |
| s1488 | 132 | 139 | 116 | 100 | 100 | 2 |
| s1494 | 134 | 140 | 118 | 102 | 102 | 2 |
| s1a | 57 | 89 | 51 | 49 | 49 | 2 |
| s208 | 23 | 42 | 21 | 20 | 17 | 2 |
| s27 | 10 | 22 | 10 | 10 | 10 | 1 |
| s386 | 33 | 46 | 29 | 25 | 20 | 1 |
| s420 | 29 | 50 | 28 | 27 | 27 | 4 |
| s510 | 67 | 67 | 51 | 48 | 48 | 4 |
| s820 | 13 | 13 | 13 | 14 | 14 | 1 |
| s832 | 106 | 100 | 86 | 76 | 76 | 4 |
| s840 | 98 | 97 | 80 | 72 | 65 | 4 |
| sand | 143 | 143 | 125 | 112 | 112 | 3 |
| shiftreg | 3 | 7 | 3 | 5 | 5 | 0 |
| sse | 40 | 44 | 37 | 33 | 29 | 1 |
| styr | 102 | 129 | 90 | 82 | 82 | 2 |
| tma | 52 | 46 | 46 | 40 | 34 | 2 |
| **Total** | 2099 | 2395 | 1780 | 1621 | 1533 | |
| **Percentage, %** | 136.92 | 156.23 | 116.11 | 105.74 | 100.00 | |

Furthermore, from Table 9 we see that the values of LUT counts are the same for some equivalent $P_T$ and $P_{2C}$ FSMs that do not belong to the set 0. This phenomenon occurs for the following benchmarks: *dk16, ex1, planet, planet1, s1488, s1494, s1a, s420, s510, s810, s832, sand* and *styr*. Analysis of Table 8 reveals the nature of this phenomenon: there are more than $S_L = 6$ bits in state codes for these FSMs. This means that the following condition holds:

$$R_{MB} > S_L. \tag{34}$$

In this case, the condition (14) is violated. This leads to the empty set $A_{MB}$. In turn, this makes correct the following relations: $A_{PC} = A$ and $\pi_A = \pi_V$. So, if the condition (34) holds, then $P_{2C}$ FSMs turn into $P_T$ FSMs. Obviously, there are the same LUT counts for such equivalent $P_{2C}$ and $P_T$ FSMs.

**Table 10.** Experimental results (the minimum cycle time, nanoseconds).

| Benchmark | Auto | One-Hot | JEDI | $P_T$ | Our Approach | Set |
|---|---|---|---|---|---|---|
| bbara | 8.811 | 8.811 | 8.352 | 8.394 | 7.601 | 1 |
| bbsse | 10.096 | 9.642 | 9.213 | 8.763 | 7.924 | 1 |
| bbtas | 8.497 | 8.497 | 8.451 | 8.497 | 8.497 | 0 |
| beecount | 9.605 | 9.605 | 8.941 | 8.568 | 7.740 | 1 |
| cse | 10.558 | 9.840 | 9.343 | 8.570 | 7.764 | 1 |
| dk14 | 8.821 | 9.395 | 8.762 | 8.964 | 8.070 | 1 |
| dk15 | 8.797 | 8.998 | 8.735 | 8.890 | 8.009 | 1 |
| dk16 | 9.491 | 9.320 | 8.672 | 8.327 | 7.539 | 1 |
| dk17 | 8.617 | 9.587 | 8.617 | 8.617 | 8.602 | 0 |
| dk27 | 8.325 | 8.424 | 8.369 | 8.325 | 8.325 | 0 |
| dk512 | 8.566 | 8.566 | 8.477 | 8.566 | 8.566 | 0 |
| donfile | 9.033 | 9.034 | 8.509 | 7.916 | 7.628 | 1 |
| ex1 | 10.425 | 10.955 | 9.454 | 8.496 | 8.496 | 2 |
| ex2 | 8.635 | 8.635 | 8.596 | 8.566 | 7.738 | 1 |
| ex3 | 8.731 | 8.731 | 8.707 | 8.731 | 8.731 | 0 |
| ex4 | 9.214 | 9.315 | 8.874 | 8.745 | 7.902 | 1 |
| ex5 | 9.147 | 9.147 | 9.119 | 9.147 | 9.147 | 0 |
| ex6 | 9.564 | 9.772 | 9.330 | 8.701 | 7.863 | 1 |
| ex7 | 8.598 | 8.578 | 8.584 | 8.582 | 7.751 | 1 |
| keyb | 10.121 | 10.699 | 9.666 | 9.063 | 8.174 | 1 |
| kirkman | 10.971 | 10.392 | 10.280 | 9.621 | 8.300 | 2 |
| lion | 8.539 | 8.501 | 8.541 | 8.595 | 8.595 | 0 |
| lion9 | 8.470 | 8.998 | 8.444 | 8.427 | 8.427 | 0 |
| mark1 | 9.825 | 9.825 | 9.343 | 8.942 | 8.063 | 1 |
| mc | 8.688 | 8.719 | 8.682 | 8.688 | 8.688 | 0 |
| modulo12 | 8.302 | 8.302 | 8.299 | 8.302 | 8.302 | 0 |
| opus | 9.684 | 9.684 | 9.275 | 9.290 | 8.353 | 1 |
| planet | 11.264 | 11.264 | 9.073 | 8.897 | 8.897 | 2 |
| planet1 | 11.264 | 11.264 | 9.073 | 8.897 | 8.897 | 2 |
| pma | 10.634 | 10.634 | 9.681 | 9.215 | 7.963 | 2 |
| s1 | 10.623 | 11.154 | 10.156 | 9.669 | 8.308 | 2 |
| s1488 | 11.013 | 11.372 | 10.155 | 9.114 | 9.114 | 2 |
| s1494 | 10.487 | 10.654 | 9.878 | 9.163 | 9.163 | 2 |
| s1a | 10.313 | 9.462 | 9.704 | 9.385 | 9.385 | 2 |
| s208 | 9.503 | 9.434 | 9.361 | 8.859 | 7.684 | 2 |
| s27 | 8.672 | 8.862 | 8.662 | 8.671 | 7.832 | 1 |
| s386 | 9.676 | 9.494 | 9.311 | 9.205 | 8.298 | 1 |
| s420 | 9.864 | 9.780 | 9.755 | 9.619 | 9.619 | 4 |
| s510 | 9.742 | 9.742 | 9.155 | 8.889 | 8.889 | 4 |
| s820 | 10.691 | 10.641 | 9.775 | 9.317 | 9.317 | 1 |
| s832 | 10.975 | 10.638 | 9.866 | 9.297 | 9.297 | 4 |
| s840 | 9.195 | 9.228 | 9.158 | 9.248 | 8.321 | 4 |
| sand | 12.390 | 12.390 | 11.652 | 9.895 | 9.895 | 3 |
| shiftreg | 8.302 | 7.265 | 7.091 | 8.302 | 8.302 | 0 |
| sse | 10.096 | 9.642 | 9.455 | 9.002 | 8.597 | 1 |
| styr | 11.067 | 11.497 | 10.666 | 9.398 | 9.398 | 2 |
| tma | 9.831 | 10.495 | 9.821 | 9.247 | 7.974 | 2 |
| **Total** | 453.73 | 454.88 | 431.08 | 417.58 | 395.94 | |
| **Percentage, %** | 114.60 | 114.89 | 108.88 | 105.46 | 100.00 | |

As follows from Table 10, the circuits of $P_{2C}$-based FSMs are the fastest compared to the circuits produced by other investigated methods. There is the following gain: (1) 14.60%

compared to Auto-based FSMs; (2) 14.89% compared to One-hot–based FSMs; (3) 8.88% compared to JEDI-based FSMs; (4) 5.46% compared to $P_T$-based FSMs. We think that this gain is due to the fact that in some cases the circuits of $P_{2C}$-based FSMs have fewer levels of LUTs than the circuits of $P_T$-based FSMs. We discussed the reasons for this phenomenon in the analysis of Table 9. It is interesting to note that the average gain in the cycle time almost coincides with the average gain in the LUT counts (for $P_T$- and $P_{2C}$-based FSMs).

**Table 11.** Experimental results (the maximum operating frequency, MHz).

| Benchmark | Auto | One-Hot | JEDI | $P_T$ | Our Approach | Set |
|---|---|---|---|---|---|---|
| Benchmark | Auto | One-Hot | JEDI | PT FSM | PE FSM | Set |
| bbara | 113.496 | 113.496 | 119.727 | 119.139 | 131.556 | 1 |
| bbsse | 99.049 | 103.713 | 108.539 | 114.116 | 126.199 | 1 |
| bbtas | 117.687 | 117.687 | 118.336 | 117.687 | 117.687 | 0 |
| beecount | 104.112 | 104.112 | 111.839 | 116.720 | 129.199 | 1 |
| cse | 94.713 | 101.626 | 107.030 | 116.680 | 128.807 | 1 |
| dk14 | 113.364 | 106.439 | 114.134 | 111.556 | 123.908 | 1 |
| dk15 | 113.675 | 111.137 | 114.487 | 112.485 | 124.862 | 1 |
| dk16 | 105.362 | 107.294 | 115.316 | 120.096 | 132.647 | 1 |
| dk17 | 116.049 | 104.308 | 116.049 | 116.049 | 116.249 | 0 |
| dk27 | 120.122 | 118.709 | 119.494 | 120.122 | 120.122 | 0 |
| dk512 | 116.740 | 116.740 | 117.963 | 116.740 | 116.740 | 0 |
| donfile | 110.706 | 110.696 | 117.517 | 126.323 | 131.093 | 1 |
| ex1 | 95.922 | 91.281 | 105.777 | 117.700 | 117.700 | 2 |
| ex2 | 115.808 | 115.808 | 116.340 | 116.744 | 129.234 | 1 |
| ex3 | 114.536 | 114.536 | 114.846 | 114.536 | 114.536 | 0 |
| ex4 | 108.530 | 107.352 | 112.690 | 114.356 | 126.552 | 1 |
| ex5 | 109.327 | 109.327 | 109.661 | 109.327 | 109.327 | 0 |
| ex6 | 104.556 | 102.333 | 107.183 | 114.930 | 127.186 | 1 |
| ex7 | 116.306 | 116.576 | 116.495 | 116.526 | 129.011 | 1 |
| keyb | 98.806 | 93.466 | 103.453 | 110.340 | 122.342 | 1 |
| kirkman | 91.148 | 96.232 | 97.272 | 103.938 | 120.476 | 2 |
| lion | 117.110 | 117.634 | 117.083 | 116.353 | 116.353 | 0 |
| lion9 | 118.065 | 111.136 | 118.421 | 118.668 | 118.668 | 0 |
| mark1 | 101.781 | 101.781 | 107.032 | 111.834 | 124.020 | 1 |
| mc | 115.102 | 114.694 | 115.174 | 115.102 | 115.102 | 0 |
| modulo12 | 120.454 | 120.454 | 120.498 | 120.454 | 120.454 | 0 |
| opus | 103.265 | 103.265 | 107.818 | 107.642 | 119.717 | 1 |
| planet | 88.777 | 88.777 | 110.222 | 112.395 | 112.395 | 2 |
| planet1 | 88.777 | 88.777 | 110.222 | 112.395 | 112.395 | 2 |
| pma | 94.039 | 94.039 | 103.293 | 108.524 | 125.587 | 2 |
| s1 | 94.134 | 89.653 | 98.465 | 103.426 | 120.362 | 2 |
| s1488 | 90.800 | 87.934 | 98.472 | 109.727 | 109.727 | 2 |
| s1494 | 95.357 | 93.861 | 101.236 | 109.135 | 109.135 | 2 |
| s1a | 96.963 | 105.687 | 103.048 | 106.558 | 106.558 | 2 |
| s208 | 105.231 | 106.000 | 106.825 | 112.874 | 130.136 | 2 |
| s27 | 115.314 | 112.842 | 115.449 | 115.324 | 127.676 | 1 |
| s386 | 103.348 | 105.329 | 107.401 | 108.642 | 120.512 | 1 |
| s420 | 101.378 | 102.249 | 102.514 | 103.961 | 103.961 | 4 |
| s510 | 102.648 | 102.648 | 109.226 | 112.493 | 112.493 | 4 |
| s820 | 93.537 | 93.975 | 102.300 | 107.336 | 107.336 | 1 |
| s832 | 91.117 | 94.001 | 101.354 | 107.563 | 107.563 | 4 |
| s840 | 108.755 | 108.364 | 109.196 | 108.133 | 120.184 | 4 |
| sand | 80.711 | 80.711 | 85.821 | 101.059 | 101.059 | 3 |
| shiftreg | 120.454 | 137.645 | 141.028 | 120.454 | 120.454 | 0 |
| sse | 99.049 | 103.713 | 105.760 | 111.085 | 116.315 | 1 |
| styr | 90.359 | 86.979 | 93.754 | 106.411 | 106.411 | 2 |
| tma | 101.719 | 95.284 | 101.819 | 108.141 | 125.413 | 2 |
| **Total** | 4918.26 | 4910.30 | 5157.58 | 5301.80 | 5605.42 | |
| **Percentage, %** | 87.74 | 87.60 | 92.01 | 94.58 | 100.00 | |

As follows from Table 10, for the Set 0, there are the same values of cycle times for equivalent benchmarks using models of single-core and dual-core FSMs. The explanation is the same as it is for the equality of LUT counts. Moreover, from Table 10 we can find out that the temporal characteristics are the same for the following benchmarks: *dk16*, *ex1*, *planet*, *planet1*, *s1488*, *s1494*, *s1a*, *s420*, *s510*, *s810*, *s832*, *sand* and *styr*. The reasons for this phenomenon have also been analyzed in the previous paragraphs.

Using values of cycle times, we can trivially compute the values of maximum operating frequencies. These values are shown in Table 11.

As follows from Table 11, the circuits of $P_{2C}$-based FSMs have the highest values of maximum operating frequencies compared to the circuits based on other investigated methods. There is the following gain: (1) 12.26% compared to Auto-based FSMs; (2) 12.40% compared to One-hot–based FSMs; (3) 7.09% compared to JEDI-based FSMs; (4) 5.42% compared to equivalent $P_T$-based FSMs. Obviously, the gain in frequency is related to the gain in cycle time. We discussed all the reasons for this phenomenon above.

The value of power consumption is one of the most important characteristics of FSM circuits [44]. Very often, the gain in area-temporal characteristics is accompanied with an increase in the power consumption [27]. Using Vivado reports allows constructing Table 12 with values of consumed power.

The main goal of the proposed method is to obtain FSM circuits with fewer LUTs than it is in circuits of equivalent $P_T$-based FSMs. Of course, this improvement can lead to an increase in power consumption. As follows from Table 12, this increase is extremely small. Compared to $P_T$-based FSMs, the circuits of equivalent $P_{2C}$-based FSMs consume less than one percent more power (0.76%). If compare $P_{2C}$-based FSMs with other investigated methods, then there is the following gain: (1) 16.38% compared to Auto-based FSMs; (2) 24.02% compared to One-hot–based FSMs; (3) 1.90% compared to JEDI-based FSMs.

We associate this loss with the following. In $P_T$-based FSMs, the state variables $T_r \in T$ are connected only with the block *LUTerV*. However, in $P_{2C}$-based FSMs, these variables are connected with LUTs of both *LUTerV* and *CoreT*. This increase in the number of connections leads to an increase in the value of parasitic capacitance in an FSM circuit [26]. Due to it, $P_{2C}$-based FSMs consume more power than equivalent $P_T$-based FSMs. Obviously, this phenomenon does not occur for FSMs from the Set 0. Moreover, for the benchmarks *dk16*, *ex1*, *planet*, *planet1*, *s1488*, *s1494*, *s1a*, *s420*, *s510*, *s810*, *s832*, *sand* and *styr* both $P_T$- and $P_{2C}$-based FSMs consume equal values of power.

So, the proposed method allows obtaining circuits with either better or the same values of area-temporal characteristics than they are for equivalent $P_T$-based FSMs. Our main purpose is to get the FSM circuits with fewer LUTs than it is for equivalent $P_T$-based FSMs. As follows from the conducted experiments, this goal has been achieved. Furthermore, the proposed method has an additional positive effect: it allows getting faster FSM circuits than the circuits of equivalent $P_T$-based FSMs. Our method loses slightly in terms of the amount of power consumed. However, this loss does not exceed 1% on average. We think that our approach can be used instead of $P_T$ FSMs if area-temporal characteristics determine the optimality of the resulting FSM circuits.

**Table 12.** Experimental results (the consumed power, Watts).

| Benchmark | Auto | One-Hot | JEDI | $P_T$ | Our Approach | Set |
|---|---|---|---|---|---|---|
| bbara | 0.961 | 0.961 | 0.880 | 0.818 | 0.841 | 1 |
| bbsse | 2.651 | 1.637 | 2.144 | 2.028 | 2.072 | 1 |
| bbtas | 0.900 | 0.900 | 0.900 | 0.900 | 0.900 | 0 |
| beecount | 2.011 | 2.011 | 1.401 | 1.389 | 1.392 | 1 |
| cse | 1.389 | 1.450 | 1.322 | 1.306 | 1.312 | 1 |
| dk14 | 3.339 | 3.710 | 3.332 | 3.301 | 3.321 | 1 |
| dk15 | 1.783 | 2.285 | 1.779 | 1.712 | 1.728 | 1 |
| dk16 | 3.334 | 3.109 | 2.879 | 2.801 | 2.801 | 1 |
| dk17 | 2.268 | 2.302 | 2.258 | 2.286 | 2.286 | 0 |
| dk27 | 1.524 | 1.210 | 1.514 | 1.539 | 1.539 | 0 |
| dk512 | 1.852 | 1.852 | 1.701 | 1.743 | 1.743 | 0 |
| donfile | 1.076 | 1.076 | 0.970 | 0.912 | 0.934 | 1 |
| ex1 | 4.564 | 3.430 | 2.804 | 2.612 | 2.612 | 2 |
| ex2 | 0.735 | 0.753 | 0.709 | 0.698 | 0.712 | 1 |
| ex3 | 0.758 | 0.758 | 0.758 | 0.758 | 0.758 | 0 |
| ex4 | 1.980 | 1.659 | 1.605 | 1.589 | 1.605 | 1 |
| ex5 | 0.754 | 0.754 | 0.752 | 0.765 | 0.765 | 0 |
| ex6 | 2.675 | 4.256 | 2.648 | 2.613 | 2.661 | 1 |
| ex7 | 1.359 | 1.548 | 1.361 | 1.342 | 1.392 | 1 |
| keyb | 1.524 | 1.502 | 1.506 | 1.492 | 1.501 | 1 |
| kirkman | 2.204 | 2.355 | 1.950 | 1.846 | 1.852 | 2 |
| lion | 0.909 | 0.996 | 0.914 | 0.923 | 0.923 | 0 |
| lion9 | 1.100 | 1.337 | 1.095 | 1.102 | 1.102 | 0 |
| mark1 | 1.851 | 1.851 | 1.633 | 1.621 | 1.643 | 1 |
| mc | 0.827 | 0.941 | 0.823 | 0.823 | 0.823 | 0 |
| modulo12 | 0.915 | 0.915 | 0.919 | 0.921 | 0.921 | 0 |
| opus | 1.750 | 1.750 | 1.689 | 1.678 | 1.714 | 1 |
| planet | 4.553 | 4.553 | 2.887 | 2.714 | 2.714 | 2 |
| planet1 | 4.553 | 4.553 | 2.887 | 2.714 | 2.714 | 2 |
| pma | 1.818 | 1.818 | 1.701 | 1.686 | 1.717 | 2 |
| s1 | 3.133 | 3.578 | 2.966 | 2.895 | 2.918 | 2 |
| s1488 | 4.430 | 4.544 | 3.996 | 3.801 | 3.801 | 2 |
| s1494 | 3.527 | 3.626 | 3.430 | 3.396 | 3.396 | 2 |
| s1a | 1.770 | 2.458 | 1.656 | 1.602 | 1.602 | 2 |
| s208 | 1.858 | 3.311 | 1.740 | 1.694 | 1.726 | 2 |
| s27 | 1.148 | 2.342 | 1.157 | 1.114 | 1.143 | 1 |
| s386 | 1.682 | 1.824 | 1.552 | 1.501 | 1.543 | 1 |
| s420 | 1.960 | 3.443 | 1.909 | 1.812 | 1.812 | 4 |
| s510 | 2.166 | 2.166 | 1.714 | 1.643 | 1.643 | 4 |
| s820 | 1.128 | 1.197 | 1.124 | 1.112 | 1.112 | 1 |
| s832 | 2.662 | 2.409 | 2.071 | 1.985 | 1.985 | 4 |
| s840 | 2.704 | 2.695 | 2.436 | 2.243 | 2.315 | 4 |
| sand | 1.640 | 1.640 | 1.479 | 1.401 | 1.401 | 3 |
| shiftreg | 0.879 | 0.959 | 0.868 | 0.879 | 0.879 | 0 |
| sse | 1.651 | 1.727 | 1.520 | 1.503 | 1.521 | 1 |
| styr | 4.506 | 5.233 | 3.649 | 3.598 | 3.598 | 2 |
| tma | 2.020 | 1.745 | 1.752 | 1.711 | 1.763 | 2 |
| **Total** | 96.78 | 103.13 | 84.74 | 82.52 | 83.162 | |
| **Percentage, %** | 116.38 | 124.02 | 101.90 | 99.24 | 100.00 | |

## 8. Conclusions

Modern FPGAs are very powerful design tools [45]. Nowadays, a single FPGA chip may implement a very complicated digital system. The main drawback of FPGAs is a very small number of LUT inputs [19,46]. This complicates the problem of optimizing the FSM circuits representing sequential blocks of digital systems. Very often, the process of technology mapping for such FSMs is connected with applying various functional decomposition methods. In this case, the resulting LUT-based FSM circuits are multi-level.

The technology mapping can be based on applying various methods of structural decomposition [30]. The research results shown in [11] prove that, very often, the SD leads to FSM circuits with significantly better characteristics compared to their counterparts based on the FD. Our research [12] shows that single-core circuits with the twofold state assignment have better characteristics compared to their FD-based counterparts. However, this approach is connected with using a special transformer creating the extended state codes. This transformer consumes some resources of FPGA chip used.

In our current article, we propose to use two cores generating systems of partial Boolean functions. This leads to $P_{2C}$ Mealy FSMs where different systems of state variables are used in different cores. Our approach allows reducing LUT counts and improving temporal characteristics in comparison with PT-based FSMs. Note that this gain is associated with a very slight increase in the power consumption (up to 1% on average).

In our future research, we will try to use this approach to optimize Mealy FSM circuits based on various structural decomposition methods. We will also check the possibility of using the double-core approach for optimizing the circuits of LUT-based Moore FSMs. We hope these methods can be used for implementing sequential devices of modern embedded systems.

**Author Contributions:** Conceptualization, A.B., L.T. and K.K.; methodology, A.B., L.T. and K.K.; formal analysis, A.B., L.T. and K.K.; writing—original draft preparation, A.B., L.T. and K.K.; supervision, A.B. All authors have read and agreed to the published version of the manuscript.

**Funding:** This research received no external funding.

**Data Availability Statement:** The data presented in this study are available in the article.

**Conflicts of Interest:** The authors declare no conflict of interest.

## Abbreviations

The following abbreviations are used in this manuscript:

| | |
|---|---|
| AIR | additional input register |
| CAD | computer aided design |
| CLB | configurable logic block |
| ESC | extended state codes |
| FAB | function assembly block |
| FD | functional decomposition |
| FPGA | field-programmable gate array |
| FSM | finite state machine |
| IMF | input memory function |
| LUT | look-up table |
| MBC | maximum binary state codes |
| PBF | partial Boolean functions |
| SBF | system of Boolean functions |
| SD | structural decomposition |
| SOP | sum of products |
| STG | state transitions graph |
| STT | state transition table |
| TSA | twofold state assignment |

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
