# Peer review of "Using a Double-Core Structure to Reduce the LUT Count in FPGA-Based Mealy FSMs"

_electronics, doi:10.3390/electronics11193089_

Round 1

Reviewer 1 Report

The article sets out the main goal of the work from the very beginning, which is: "design method aimed at reducing the LUT counts in the circuits of FPGA-based Mealy FSMs with twofold state assignment".

The article is written in a very legible way, despite the complicated issue. Each subsequent paragraph and chapter create a coherent whole. The authors put a lot of work into preparing a clear article. I found only a few small things that need to be considered and improved.

All mentioned comments are only notes to the editorial part:

- figure 1 and line 83 - the names LuterT and LuterY are used in the figure and LUTerT is used on line 83. It is worth standardizing it

- graphic 5 and table 1 - there is a discrepancy between graphic 5 and table 1 for:

-> h = 2 - Xh value is incorrect (/ X1 or / X2)

-> h = 7 - Xh value is incorrect (/ X1 or X2)

-> h = 8 - the value of Yh (first) is incorrect (Y1 or Y2)

Therefore, please also check the text below the table referring to it.

Finally, it is worth noting that the authors conducted a number of experiments which they critically assessed when drawing conclusions from them. They tested their theory not on one example benchmark, but on many. This is very valuable, and I believe it is quite a contribution to the research work.

Author Response

Dear Sir or Madame!

Thank you for your valuable time spent reviewing our article! Thank you for the high evaluation of our results and comments. Please, find in the attachment our answers concerning your remarks. In the pdf document all essential changes are highlighted.

Yours sincerely,

Authors

Reviewer 2 Report

Manuscript is solid, my minor concern being wrt English. Checking the paper once again would be useful for avoiding parts like: "In this paper, there is proposed a method for synthesis of LUT" etc. 

Author Response

Dear Sir or Madame!

Thank you for your valuable time spent reviewing our article! Thank you for the high evaluation of our results!

We analyzed the text and changed the sentences that used the introductory part: ”In this paper”. We hope that now the text looks better without these repetitions. In the pdf document all essential changes are highlighted.

                     Thank you very much once more!

                                            Kind regards.

                                             Authors

Reviewer 3 Report

1.      What are the main questions that remained to be addressed by the authors of the article? 

There are no MAIN questions opened because the proposed work is, in my opinion, complete and solid. For a better understanding of the material, I would suggest the Authors address these MINOR points:

A.      The use of a transformer creating extended states code is one of the key feature of the proposed methodology. This implementation consumes some chip resources: what about the increase of energy consumption in the overall energy budget of the chip ? The Authors are invited to add few comments on this aspect.

Author Response

(The authors gave the same response as above.)
